# Study on the Evolution of Physical Parameters and Dynamic Compression Mechanical Properties of Granite after Different Heating and Cooling Cycles

**DOI:** 10.3390/ma16062300

**Published:** 2023-03-13

**Authors:** Hongzhong Zhang, Linqi Huang, Xibing Li, Xingmiao Hu, Yangchun Wu

**Affiliations:** School of Resources and Safety Engineering, Central South University, Changsha 410083, China; zhz0804@csu.edu.cn (H.Z.); xbli@csu.edu.cn (X.L.); huxingmiao@csu.edu.cn (X.H.); wuyangchun1995@csu.edu.cn (Y.W.)

**Keywords:** heating-cooling cycle, cooling mode, physical parameters, microscopic damage, dynamic compression properties

## Abstract

The study of the evolution law of basic physical parameters and dynamic compression performance of deep granite under the environment of the heating-cooling cycle is of great significance for the stability evaluation of deep underground engineering and the development of deep resources. In this study, heating-cooling cycle tests and dynamic compression tests were conducted on a large number of fine-grained granite specimens with heating temperatures from 200 to 600 °C and times from one to twenty times using a box-type high-temperature muffle furnace and Hopkinson pressure bar (SHPB) test system, and the evolution law of basic physical parameters and dynamic compression mechanical properties of fine-grained granite were studied using theoretical and fitting analysis. The test results showed that: the changes of the basic physical parameters of granite have obvious temperature effect; 600 °C is a threshold value for the changes of each physical parameter of granite; the sensitivity of each physical parameter to the number of heating and cooling cycles is small before 600 °C; and the sensitivity of each physical parameter to the number of heating and cooling cycles significantly increases at 600 °C. The dynamic compressive strength and elastic modulus of granite decreased with the increase in heating and cooling cycles, and the maximum decrease rate was 89.1% and 85.9%, respectively, and the strain rate linearly increased with the increase in heating and cooling cycles, and the maximum strain rate was 123 s^−1^. The temperature, the number of heating and cooling cycles, and the impact air pressure, all had significant effects on the damage mode and crushing degree of granite.

## 1. Introduction

With the construction of deep underground engineering (nuclear waste storage), the mining of dry hot rock (Figure 1) and the development of deep mining of metal mineral resources, the mechanical properties and stability of rock masses are significantly influenced by the deep water-heat alternating action environment, and the evolution law of basic physical parameters and dynamic properties of deep rock masses under the heating-cooling cycle action environment have become a hot research topic in recent years in rock mechanics. The study of the evolution of basic physical parameters and dynamic compression properties of rocks after the heating-cooling cycle is important for the drilling and mining of dry hot rock, which can effectively improve the mining efficiency of dry hot rock; on the other hand, it can provide a theoretical basis for the development of the composite rock-breaking method of the heating-cooling cycle and mechanical impact, as well as high-temperature solution leaching mining.

Many scholars at home and abroad have studied the effect of temperature on granite from various directions and have achieved a lot of results [1,2]. Li et al. [3], Wang et al. [4], and Zhao et al. [5] studied the effect of different heat treatments on the fracture characteristics of granite. Yin et al. [6,7] and Li et al. [8] studied the effect of temperature on the dynamic mechanical properties of granite. Sun et al. [9] studied the effect of temperature on the physical properties of granite such as quality and surface color. Xu et al. [10] and Zhou et al. [11] investigated the thermal microcrack extension of granite using optical microscopy, thin section observation and scanning electron microscopy (SEM). Zhang et al. [12] and Hao et al. [13] investigated the effect of temperature on the physical and mechanical properties, as well as the intrinsic structure relationship, of granite. Ding et al. [14] studied the effect of temperature on the permeability of fractured granite. Wang et al. [15,16] investigated the effect of two different heating methods, slow heating and fast heating, on the thermal cracking of granite.

Many others have studied the effect of different cooling methods on the mechanical properties of granite. Xi et al. [17] studied the effects of natural cooling, water cooling and liquid nitrogen (LN2) cooling on the dynamic mechanical properties of rocks and their fracture characteristics. Zhao et al. [18] studied the effect of different temperatures and cooling methods on parameters such as surface hardness and the uniaxial compressive strength of granite. Tang et al. [19] investigated the effect of temperature on the basic friction angle of granite cracks under water and liquid nitrogen cooling methods, different specimen sizes and tilt rates. Shao et al. [20] and Chen et al. [21] investigated the physical and mechanical damage characteristics of granite at different high temperatures under the effect of liquid nitrogen cooling. Guan et al. [22] studied the damage characteristics of granite specimens and their mechanisms after different heating-cooling cycles using air-cooling and liquid nitrogen cooling.

The physical and mechanical properties of granite under the action of heating and cooling cycles have also been studied. Yin et al. [23] and Li et al. [24] studied the effects of cyclic heating and cooling treatments on the microscopic pore structure of granite. Chen et al. [25] studied the effects of combined cooling on the mechanical properties of granite at different temperatures and numbers of hot and cold cycles. Zhu et al. [26] studied the effects of longitudinal wave velocity, uniaxial compressive strength, and elastic modulus of granite specimens to different cyclic heating and water cooling treatments. Wu et al. [27] and Wang et al. [28] studied the physical and mechanical properties and crack expansion of granite under single and multiple heating-cooling cycles. Yin et al. [29] investigated the microscopic thermal damage mechanism and mechanical properties of granite under cyclic heating-cooling. Zhang et al. [30] developed a method to realize the mechanical properties of granite in deep ground and established a numerical simulation method that enables long-term mechanical analysis of reservoir rocks considering thermal damage in deep geothermal engineering. Zhu et al. [31] analyzed changes in the mechanical properties of granite after various thermal impacts. Sun et al. [32] investigated the effect of cyclic heating and cooling effects on the thermal diffusion coefficient of granite at different temperatures. Ge et al. [33] and Jiang et al. [34] studied the effect of heating, and Li et al. [35] studied the effect of different cooling methods on the dynamic mechanical properties of granite. 

From the above analysis, it can be seen that there have been relatively few studies on the dynamic compressive properties and damage characteristics of granite after the cyclic action of heating and cooling at different temperatures by domestic and foreign scholars, and most of the studies on the effect of temperature on the physical and mechanical properties of granite are single cooling action or using a single cooling mode, there are few studies on the physical and mechanical properties of granite after heating and cooling cycles under multiple cooling methods. In this study, the evolution law of basic physical parameters and dynamic compression mechanical properties and damage characteristics of granite after cyclic heating and cooling are analyzed by using water cooling and natural cooling, and the physical and mechanical properties of granite under heating and cooling cycles are further studied, which is important for improving the reservoir modification and drilling efficiency of dry hot rocks.

In this study, many fine-grained granite specimens were subjected to heating-cooling cycles and dynamic impact tests from 200 to 600 °C one to twenty times using a box-type high-temperature muffle furnace and SHPB test system, and the evolution of the basic physical parameters such as mass and volume of fine-grained granite under the effect of heating-cooling cycles and dynamic compression mechanical properties were summarized.

## 2. Specimen Preparation and Experimental Process

### 2.1. Preparation of Rock Specimens

The test material selected for this test was fine-grained granite from Zhangzhou City, Fujian Province, China, and the specimens were processed from the same granite raw material, which had a dense structure and high strength, and could be better processed for heating and cooling cycles. The specimens were processed in strict accordance with the requirements of T/CSRME 001-2019 Rock Dynamic Properties Test Procedure [36]. It was required that the non-parallelism and non-perpendicularity of each specimen were less than 0.02 mm, and the error of specimen height and diameter was less than 0.3 mm, and the specimen was processed into a cylindrical rock specimen with the specification of φ50 mm × 50 mm (Figure 2).

### 2.2. Experimental Process

The whole test process is divided into four steps: measurement of physical parameters before treatment, heating and cooling cycle treatment, measurement of physical parameters after treatment and dynamic compression; the test process and equipment used are shown in Figure 3.

(1) Measurement of physical parameters before treatment. The physical quantities to be measured are mass, volume, density, longitudinal wave velocity and porosity of the specimen, which are measured by high precision electronic scales, electronic vernier calipers and HS-YS4A Rock Acoustic Wave Parameter Tester.

(2) Heating and cooling cycle treatment. Heating and cooling cycle treatment mainly has two kinds of fast water cooling and natural cooling; the heating temperature is 200 °C, 400 °C and 600 °C, and the number of cycles is 1, 5, 10, 20 times, respectively. Firstly, put the completed numbered specimens into the high temperature muffle furnace, heat them to the specified temperature at a heating rate of 5 °C/min, keep them warm for 2 h so that the specimens are evenly heated, and then quickly use the corresponding cooling method (fast water cooling, natural cooling) to fully cool them for 2 h after the heating is completed, and repeat the above steps until the heating and cooling treatment is completed for 5, 10 and 20 times.

(3) Measure the relevant physical parameters of the specimens treated by the heating and cooling cycle again, analyze their change characteristics and summarize the influence law of heating and cooling cycle effects on the physical parameters of the granite.

(4) The specimens after the heating and cooling cycle were subjected to uniaxial dynamic compression experiments using a Hopkinson pressure bar (SHPB) test system with an impact air pressure of 0.6, 0.7 and 0.8 MPa to analyze the effect of the heating and cooling cycle on the dynamic compression performance of the granite.

### 2.3. Introduction of SHPB Test System

As shown in Figure 3, the SHPB test system consists of a dynamic loading system, a strain acquisition system, a data processing system, etc. The dynamic loading system consists of cylinders, spindle-shaped bullets, an incidence bar, a transmission bar, an absorption bar, etc. The high-pressure nitrogen gas is used as the power source to push the bullets by instantaneous release, so that the bullets can obtain high speed and then achieve dynamic loading on the specimen.

## 3. Evolutionary Patterns of Basic Physical Parameters of Granite

In order to be able to better describe the effect of the heating and cooling cycle on the basic physical parameters of the granite, the physical parameters obtained from the test are uniformly processed in the form of rate of change; the specific calculation method is shown in Equation (1), and the average value of the rate of change of each physical parameter of granite after different heating and cooling cycle treatments is shown in Table 1.
(1)$$\delta_{x}=\frac{x_{b}-x_{a}}{x_{b}}\times 100\%$$
where $\delta_{x}$ is the change rate of a physical parameter, $x_{b}$ is the value of the parameter before the test treatment and $x_{a}$ is the value of the parameter after the test treatment.

### 3.1. Mass Variation Pattern

The variation relationship between the rate of change of granite mass and the number of heating and cooling cycles is shown in Figure 4.

The effect of heating and cooling cycles on the quality of the rock is mainly reflected in the loss of bound water and volatile substances inside the rock specimen at high temperature, which leads to the reduction in the quality of the rock specimen. As can be seen from Figure 4, when water cooling, the granite mass loss rate and the number of heating and cooling cycles show a linear relationship, and the slope of the straight line increases with the increase in heating temperature, indicating that the higher the temperature, the greater the rate of change in granite mass loss; when natural cooling, the granite mass loss rate and the number of heating and cooling cycles show a non-linear relationship, and the growth process with the increase of heating and cooling cycles can be divided into rapid growth (1–5), slow growth (5–10) and basic stability (10–20). From Figure 4a, it can be seen that at 200 °C and 400 °C, the change in the granite mass loss rate with the number of heating-cooling cycles is small, and the growth of the mass loss rate is only 0.03% and 0.07%, and at 600 °C, the influence of the number of heating-cooling cycles on the quality of granite is significantly enhanced, and its mass loss rate increases from 0.12% to 1.26%, and the mass loss rate increases by more than 10 times, which shows that 600 °C is an important temperature threshold for granite quality change. The same conclusion was reached by Li et al. [8], who concluded that 600 °C is the turning temperature for property degradation of granite specimens, mainly because of the α/β phase transition of quartz is around 500 °C. From Figure 5, it is also clear that the mass loss rate when water cooling under the same conditions is higher than that when natural cooling. The fitted relationship between the mass loss rate and the number of heating-cooling cycles at different temperatures is shown in Equation (2).
(2)$$\left\{\begin{array}{c} \begin{array}{c} \delta_{m}{}_{\left(T=200^{{}^{\circ}}C,W\right)}=0.0849+0.0017N\;R^{2}=0.9538\\ \delta_{m}{}_{\left(T=400{}^{{}^{\circ}}C,W\right)}=0.0975+0.0036N\;R^{2}=0.7759\\ \delta_{m}{}_{\left(T=600{}^{{}^{\circ}}C,W\right)}=0.0175+0.0604N\;R^{2}=0.9803 \end{array}\\ \begin{array}{c} \delta_{m}{}_{\left(T=200{}^{{}^{\circ}}C,NC\right)}=0.0439N^{0.2942}\;R^{2}=0.9803\\ \delta_{m}{}_{\left(T=400{}^{{}^{\circ}}C,NC\right)}=0.0527N^{0.3503}\;R^{2}=0.8196\\ \delta_{m}{}_{\left(T=600{}^{{}^{\circ}}C,NC\right)}=0.1735N^{0.0645}\;R^{2}=0.9478 \end{array} \end{array}\right.$$
where: $\delta_{m}$ is the mass change rate, %; *T* is the heating temperature, °C; *W* is the water cooling; *NC* is the natural cooling; *N* is the number of heating and cooling cycles.

### 3.2. Density Variation Pattern

The variation relationship between the rate of change of granite density and the number of heating and cooling cycles is shown in Figure 5.

Under the conditions of water cooling and natural cooling, the density decrease rate of granite and the number of heating and cooling cycles both show a linear relationship, and the slope of the straight line increases with the increase in heating temperature, indicating that the higher the temperature, the greater the rate of change in the granite density decrease rate. The density decrease of granite is mainly caused by the expansion of volume and the loss of mass during the heating process. From Figure 5a,b, the granite specimens have different sensitivities to different cooling methods, and the degree of damage of granite under water cooling is greater than that under natural cooling. At 600 °C and when the number of heating and cooling cycles increased from one to twenty times, the rate of change in density when water cooling method increased from 2.10% to 9.63%, which increased 3.59 times; while the density change rate using the natural cooling method only increased from 2.21% to 3.62%, which was only 0.64 times. At the same time, the density change has a significant temperature effect. From Figure 5a, the density change rates at a heating temperature of 600 °C are 2.10%, 3.72%, 6.88% and 9.63%, which are significantly higher than those at 200 °C (0.22%, 0.56%, 0.91% and 0.66%) and 400 °C (0.64%, 1.15%, 1.27% and 1.71%), respectively. This result is in agreement with the findings of Zhao [5], Sun [9], Shao [20] and Li [24], who found that high temperatures cause large temperature stresses inside granite specimens, resulting in severe deterioration of their physical properties. The fitted relationship between the rate of change in density and the number of heating-cooling cycles at different temperatures is shown in Equation (3).
(3)$$\left\{\begin{array}{c} \begin{array}{c} \delta_{\rho }{}_{\left(T=200{}^{{}^{\circ}}C,W\right)}=0.4049+0.0203N\;R^{2}=0.7803\\ \delta_{\rho }{}_{\left(T=400{}^{{}^{\circ}}C,W\right)}=0.7324+0.0514N\;R^{2}=0.9157\\ \delta_{\rho }{}_{\left(T=600{}^{{}^{\circ}}C,W\right)}=1.9659+0.4016N\;R^{2}=0.9681 \end{array}\\ \begin{array}{c} \delta_{\rho }{}_{\left(T=200{}^{{}^{\circ}}C,NC\right)}=0.1909+0.0235N\;R^{2}=0.8845\\ \delta_{\rho }{}_{\left(T=400{}^{{}^{\circ}}C,NC\right)}=0.5781+0.0169N\;R^{2}=0.9196\\ \delta_{\rho }{}_{\left(T=600{}^{{}^{\circ}}C,NC\right)}=2.2386+0.0759N\;R^{2}=0.8875 \end{array} \end{array}\right.$$
where: $\delta_{\rho }$ is the rate of change in density, %.

### 3.3. Volume Variation Pattern

The variation relationship between the volume expansion rate of granite and the number of heating and cooling cycles is shown in Figure 6.

From Figure 6, the volume expansion rate of granite and the number of heating and cooling cycles show a positive correlation under different cooling methods and heating temperature conditions. The whole process can be divided into three stages: one to five times for the fast rising stage; five to ten times for the slow rising stage; ten to twenty times for the basic stable stage. Under the water-cooling condition, the exponent of the multiplicative power function continuously increases with the increase in heating temperature, indicating that the higher the temperature, the more sensitive the granite volume is to the number of heating and cooling cycles when water cooling is used. Under natural cooling conditions, the sensitivity of the granite volume expansion rate to the number of heating and cooling cycles is much less than that under water cooling conditions. The heating temperature is 600 °C, the granite volume expansion rate increases from 1.99% to 9.49% when water cooling is used, which is an increase of 376.8%, while the granite volume expansion rate increases from 2.03% to 3.49% when natural cooling is used, which is only an increase of 71.9%, and the maximum volume expansion rate when water cooling is used is 2.7 times of that when the natural cooling method is used. The same conclusion was obtained by Li [8]. The fitted relationship between the volume expansion rate and the number of heating-cooling cycles at different temperatures is shown in Equation (4).
(4)$$\left\{\begin{array}{c} \begin{array}{c} \delta_{v}{}_{\left(T=200{}^{{}^{\circ}}C,W\right)}=0.2253N^{0.3356}\;R^{2}=0.6451\\ \delta_{v}{}_{\left(T=400{}^{{}^{\circ}}C,W\right)}=0.5427N^{0.3437}\;R^{2}=0.9834\\ \delta_{v}{}_{\left(T=600{}^{{}^{\circ}}C,W\right)}=1.5969N^{0.5975}\;R^{2}=0.9761 \end{array}\\ \begin{array}{c} \delta_{v}{}_{\left(T=200{}^{{}^{\circ}}C,NC\right)}=0.1629N^{0.4109}\;R^{2}=0.8149\\ \delta_{v}{}_{\left(T=400{}^{{}^{\circ}}C,NC\right)}=0.3739N^{0.3043}\;R^{2}=0.8530\\ \delta_{v}{}_{\left(T=600{}^{{}^{\circ}}C,NC\right)}=1.9053N^{0.2019}\;R^{2}=0.9065 \end{array} \end{array}\right.$$
where: $\delta_{v}$ is the rate of volume change, %.

### 3.4. Porosity Variation Pattern

The variation relationship between the rate of change of granite porosity and the number of heating and cooling cycles is shown in Figure 7.

From Figure 7a, when water cooling is used, a good linear relationship is shown between the porosity growth rate and the number of heating and cooling cycles, and the slope of the straight line increases with the increase in heating temperature; indicating that the higher the temperature, the greater the porosity growth rate with the increase in heating and cooling cycles. When the heating temperature was 200 °C and 400 °C, the level of porosity growth rate of the specimens was low, which were below 10‱. When the heating temperature increased to 600 °C, the rate of change of the porosity growth rate of the specimens with the number of heating and cooling cycles significantly increased from 7‱ to 40‱, which was an increase of 4.7 times. From Figure 7b, the porosity growth rate increases with the number of heating and cooling cycles as a multiplicative power function, and the whole process can be divided into three stages: rapid increase stage; slow increase stage; basic stability stage. The growth rate of the porosity change rate gradually decreases with the increase in heating and cooling cycles, which is due to the large number of cracks and defects generated inside the granite specimens in the first few cycles of heating and cooling cycles, and when the number of heating and cooling cycles increases again, the mineral expansion and deformation caused by its generated thermal stress has room to move, which effectively reduces the damage of thermal stress. When the heating temperature was 200 °C and 400 °C, the porosity change rate was below 4‱. When the heating temperature increased to 600 °C, the porosity growth rate of the specimen rapidly increased from 8.39‱ to 14.19‱, an increase of 0.69 times. Thus, it seems that 600 °C is an important threshold value for the porosity change of granite. The fitted relationship between the rate of change of granite porosity and the number of heating-cooling cycles at different temperatures is shown in Equation (5).
(5)$$\left\{\begin{array}{c} \delta_{p}{}_{\left(T=200{}^{{}^{\circ}}C,W\right)}=0.9399+0.1681N\;R^{2}=0.9997\\ \delta_{p}{}_{\left(T=400{}^{{}^{\circ}}C,W\right)}=2.8302+0.2775N\;R^{2}=0.9665\\ \delta_{p}{}_{\left(T=600{}^{{}^{\circ}}C,W\right)}=7.6682+1.7266N\;R^{2}=0.9503\\ \delta_{p}{}_{\left(T=200{}^{{}^{\circ}}C,NC\right)}=0.8062N^{0.3735}\;R^{2}=0.9003\\ \delta_{p}{}_{\left(T=400{}^{{}^{\circ}}C,NC\right)}=0.6280N^{0.6329}\;R^{2}=0.9968\\ \delta_{p}{}_{\left(T=600{}^{{}^{\circ}}C,NC\right)}=8.1049N^{0.1819}\;R^{2}=0.9830 \end{array}\right.$$
where: $\delta_{p}$ is the rate of change in porosity, ‱.

### 3.5. P-Wave Speed Variation Pattern

The variation relationship between the rate of change of granite wave velocity and the number of heating and cooling cycles is shown in Figure 8.

As can be seen from Figure 8, the P-wave velocity drop rate of the granite specimens gradually increased with the increase in the number of heating and cooling cycles, and the rise process can be divided into three stages: one to five times for the rapid rise stage; five to ten times for the slow rise stage; ten to twenty times for the basic stability stage. The P-wave velocity decreases the most after experiencing one to five heating and cooling cycles. This is because a large number of new cracks and defects are generated inside the granite specimen after one heating-cooling cycle action; at the same time, the volatilization of volatile substances, such as bonded water in the granite specimen under the high temperature action, leads to the increase in the volume of the original cracks and holes inside it, which finally makes the P-wave in most of the area unable to pass, so the P-wave velocity significantly drops. Meanwhile, it can be seen from Figure 8a that the temperature effect of granite wave velocity change is very significant. The number of water cooling and heating-cooling cycles is 20 times, and the wave velocity variation rate is 28% when the heating temperature is 200 °C; 67.7% when it is 400 °C; and 100% when it is 600 °C. Compared with the wave velocity variation rate at 200 °C, it increased 1.4 times and 2.6 times at 400 °C and 600 °C, respectively. Sun [9] also obtained similar conclusions in his study. Meanwhile, the wave velocity at 600 °C could not be measured due to the formation of macroscopic penetration cracks inside the granite specimens at this time, which resulted in the inability of P waves to pass through the specimens. The fitted relationship between the P-wave velocity drop rate and the number of heating-cooling cycles at different temperatures is shown in Equation (6).
(6)$$\left\{\begin{array}{c} \begin{array}{c} \delta_{wv}{}_{\left(T=200{}^{{}^{\circ}}C,W\right)}=16.4641N^{0.1630}\;R^{2}=0.9219\\ \delta_{wv}{}_{\left(T=400{}^{{}^{\circ}}C,W\right)}=44.3173N^{0.1537}\;R^{2}=0.9380\\ \delta_{wv}{}_{\left(T=600{}^{{}^{\circ}}C,W\right)}=77.7663N^{0.0898}\;R^{2}=0.9098 \end{array}\\ \begin{array}{c} \delta_{wv}{}_{\left(T=200{}^{{}^{\circ}}C,NC\right)}=10.2619N^{0.2394}\;R^{2}=0.7796\\ \delta_{wv}{}_{\left(T=400{}^{{}^{\circ}}C,NC\right)}=32.2877N^{0.1261}\;R^{2}=0.9760\\ \delta_{wv}{}_{\left(T=600{}^{{}^{\circ}}C,NC\right)}=74.9989N^{0.0459}\;R^{2}=0.9934 \end{array} \end{array}\right.$$
where: $\delta_{wv}$ is the rate of change of wave velocity, %.

### 3.6. Granite Surface Color Variation Pattern

As shown in Figure 9, the surface color of the granite specimen gradually changed from dark cyan at 25 °C to beige at 600 °C as the heating temperature increased and the number of heating and cooling cycles increased. The effect of the number of heating and cooling cycles on the color of the granite specimens was very small, and the color of the specimens was only slightly brightened when the number of heating and cooling cycles increased from one to ten times. The main reason for this phenomenon is, on the one hand, the loss of potassium feldspar from the composition of granite under high temperature conditions, which gradually lightens the color of potassium feldspar, and, on the other hand, the dehydration of water-rich dark minerals such as mica under high temperature, which lightens the color [37].

### 3.7. Granite Microstructure Variation Pattern

The changes in granite microstructure after different heating and cooling cycles are shown in Figure 10. From the SEM images in Figure 10, at room temperature (25 °C), the granite fracture surface has smooth grains and a complete structure without obvious microcracks. With the gradual increase in temperature, more along-crystal and through-crystal damage occurred in the granite section [38], which produced many cracks, and these cracks were the main cause of the deterioration of the mechanical properties of granite. Secondly, the microcracks produced inside the granite under water cooling conditions are redundant with natural cooling conditions, mainly because of the large temperature difference between the inside and outside of the granite during water cooling, the fast-cooling rate and the large thermal stress generated inside the specimen. Finally, with the increase in the number of heating and cooling cycles and the increase in temperature, the microcracks inside the specimen gradually sprouted and expanded, and finally formed macro cracks.

## 4. Dynamic Compression Properties of Granite after Heating-Cooling Cycle

### 4.1. Dynamic Stress-Strain Curve of Granite

The dynamic stress-strain curves of the granite specimens after different heating and cooling cycles are shown in Figure 11, and the dynamic mechanical parameters are shown in Table 2.

As can be seen from Figure 11, the dynamic stress-strain curves do not have an obvious compaction phase at higher impact gas pressure, which is because the closing rate of microcracks inside the specimen is much lower than the loading rate, resulting in the specimen crossing the compaction phase and reaching the elastic change phase before it can close. From Figure 11a, the stress-strain curve gradually changes from type II to type I with the increase in impact gas pressure at the same temperature, as well as the number of heating-cooling cycles. From Figure 11a,c, the stress-strain curve will gradually change from type II to type I with the increase in heating and cooling cycles at the same temperature and impact gas pressure; meanwhile, it is easier to change the stress-strain curve from type II to type I with water cooling at the same temperature and impact gas pressure. From Figure 11b,h, the stress-strain curve will gradually change from type II to type I with the increase in heating temperature under the same number of heating and cooling cycles and impact gas pressure. The experimental results were the same as those obtained by Wu [39] and Tarasov [40]. In addition, the peak stress of the specimen tends to decrease with the increase in heating temperature and the number of heating and cooling cycles at the same impact gas pressure.

### 4.2. Dynamic Compression Strength Variation Law of Granite

The relationship between the dynamic compression strength of granite specimens and the number of heating and cooling cycles are shown in Figure 12.

From Figure 12, the relationship between the dynamic compression strength of granite and the number of heating and cooling cycles satisfies the quadratic function and decreases with the increase in the number of heating and cooling cycles. When the temperature is lower than 600 °C, the dynamic compression strength of granite decreases less with the increase in heating and cooling cycles; when the temperature reaches 600 °C, the dynamic compression strength of granite sharply decreases with the increase in heating and cooling cycles, which makes the dynamic compression strength of granite rapidly decrease. The same experimental results were obtained for Li [8]. From Figure 12c, the dynamic compression strength of granite decreased by 20 MPa, 31 MPa and 131 MPa when the heating temperature was 200 °C, 400 °C and 600 °C, and the decrease rate was 7.2%, 14.0% and 89.1%, respectively. The dynamic compression strength of granite decreases 12.4 times and 6.4 times at 600 °C than at 200 °C and 400 °C, respectively, which shows that it has an obvious temperature effect. From Figure 12c,d, the dynamic compression strength of granite decays faster when the water cooling method is used. With the natural cooling method, the dynamic strength decay rate of granite is 2.6%, 1.6% and 39.0% when the heating temperature is 200 °C, 400 °C and 600 °C, respectively, and the dynamic compression strength decay rate of granite with the water-cooling method is 2.8 times, 8.8 times and 2.3 times that when using the natural cooling method, respectively.

### 4.3. Dynamic Modulus of Elastic Change Law of Granite

The relationship between the dynamic modulus of elastic of granite specimens and the number of heating and cooling cycles is shown in Figure 13.

From Figure 13a, the dynamic modulus of elastic of granite specimens and the number of heating and cooling cycles under water cooling conditions satisfy the exponential function and have a negative correlation. At 400 °C, the average dynamic modulus of granite specimens decreased from 35.7 Gpa to 22.7 Gpa, decreasing by 36.4%; at 600 °C, the average dynamic modulus of granite specimens decreased from 29.8 Gpa to 4.2 Gpa, decreasing by 85.9%. Therefore, it can be seen that the higher the temperature, the greater the change in the dynamic modulus of elasticity of the specimens, which has an obvious temperature effect. From Figure 13b, the dynamic elastic modulus of the granite specimens and the number of heating and cooling cycles under the natural cooling condition satisfy the primary function relationship. Meanwhile, with the increase in heating temperature, the more significant the effect of heating and cooling cycles on the dynamic elastic modulus of granite specimens; the absolute value of the slope of the fitted curve gradually increases from 0.72192 to 0.85842.

### 4.4. Granite Strain Rate Variation Pattern

The variation relationship between the strain rate of granite specimens and the number of heating and cooling cycles is shown in Figure 14.

As can be seen from Figure 14, the strain rates of the granite specimens all increased with the increase in the number of heating and cooling cycles. Meanwhile, the change in strain rate has a strong temperature effect. With the increase in heating temperature, the effect of the number of heating-cooling cycles on the strain rate of granite was more significant; as can be seen from Figure 14a, the slope of the fitted curve increased from 0.6419 to 1.8579 with the increase in heating temperature from 200 °C to 600 °C, an increase of 189.6%. From Figure 14a,b, it can be seen that the effect of the number of heating and cooling cycles on the strain rate of granite is more significant under the water cooling method, and the slopes of the fitted curves for water cooling are 0.6419, 0.8975 and 1.8579 for the heating temperatures of 200 °C, 400 °C and 600 °C, respectively; the slopes of the fitted curves for natural cooling are 0.2510, 0.5347 and 1.5678, corresponding to 2.56, 1.68 and 1.19 times of the slope of the water-cooled fitted curve than that of the natural-cooled fitted curve, respectively. The trend of strain rate variation is consistent with the results obtained by Zhang et al. [12].

### 4.5. Granite Dynamic Crushing Characteristics

The crushing of granite specimens under different conditions is shown in Figure 15. With the increase in heating temperature, the increase in heating and cooling cycles and the increase in impact air pressure, the degree of granite specimen crushing gradually increased and the crushed particles gradually decreased.

## 5. Conclusions

In this study, heating and cooling cycle tests were conducted on a large number of fine-grained granite specimens at temperatures of 200~600 °C between one to twenty times using a box-type high-temperature muffle furnace, and the evolution of the basic physical parameters of fine-grained granite under the effect of heating and cooling cycle was summarized; dynamic impact tests were conducted on granite specimens after the effect of the heating and cooling cycle using the SHPB test system, and their dynamic compressive mechanical properties were analyzed. 

(1) The mass, density and wave velocity of granite specimens show negative correlation with the heating temperature and the number of heating and cooling cycles, while the volume and porosity show positive correlation; the high temperature will cause a large temperature stress inside the granite specimen, so that its physical properties produce serious deterioration.

(2) The changes in the physical parameters have obvious temperature effects and are more sensitive to the water cooling method. A threshold value of 600 °C is determined for the changes in the physical parameters of granite, and the sensitivity of the physical parameters to the number of heating and cooling cycles is small before 600 °C; the sensitivity of the physical parameters to the number of heating and cooling cycles significantly increases when reaching 600 °C.

(3) With the increase in heating temperature and the increase in heating and cooling cycles, the surface color of granite specimens gradually changes from dark cyan to beige, mainly due to the loss of potassium and dehydration of water-rich dark minerals such as mica under high temperature conditions. According to SEM, it is known that the microscopic cracks of the specimens gradually sprouted and expanded until penetration, and finally formed macroscopic damage.

(4) The dynamic compression strength of granite specimens has an obvious temperature effect and a strain rate effect. Meanwhile, the number of heating and cooling cycles had some effects on the dynamic compressive strength, elastic modulus and strain rate of granite. The dynamic compressive strength and elastic modulus decreased with the increase in heating-cooling cycles, and the maximum decrease rate was 89.1% and 85.9%, and the strain rate linearly increased with the increase in heating-cooling cycles; the maximum strain rate reached 123 s^−1^.

(5) With the increase in heating temperature, the increase in heating-cooling cycles and the increase in impact air pressure, the degree of granite specimen crushing gradually increased.

(6) The study of dynamic mechanical properties of granite after heating and cooling cycles is limited to the case of uniaxial dynamic loading, and the mechanical properties under the combined dynamic and static conditions must be investigated in the future, while numerical simulations can be used to further investigate the microstructure and mechanical properties of granite after heating and cooling cycles.

## Figures and Tables

**Figure 1 materials-16-02300-f001:**
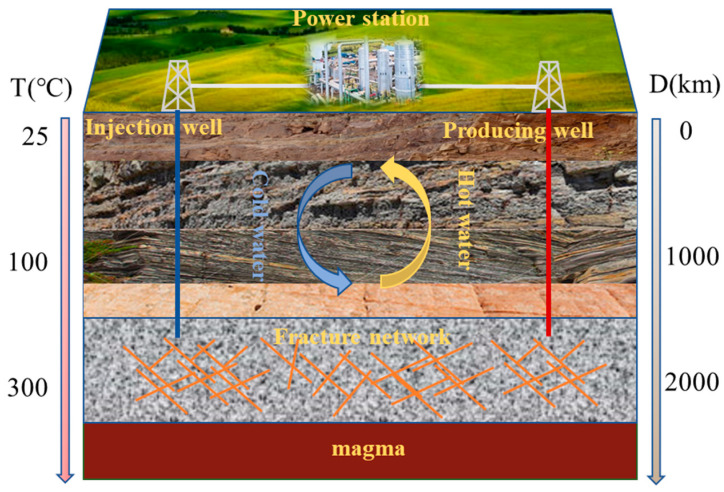
Schematic diagram of hot dry rock mining.

**Figure 2 materials-16-02300-f002:**
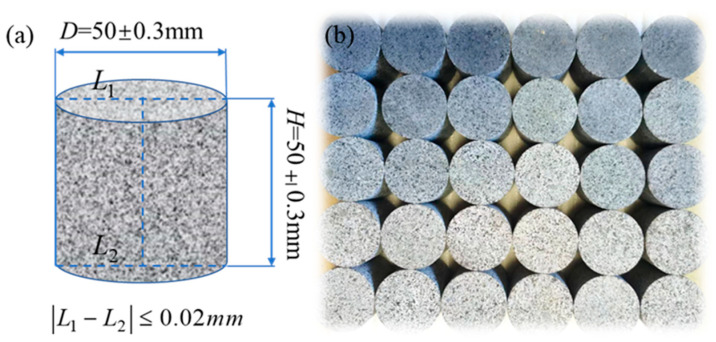
(**a**) Rock specimen processing requirements. (**b**) Partially processed fine-grained granite specimen.

**Figure 3 materials-16-02300-f003:**
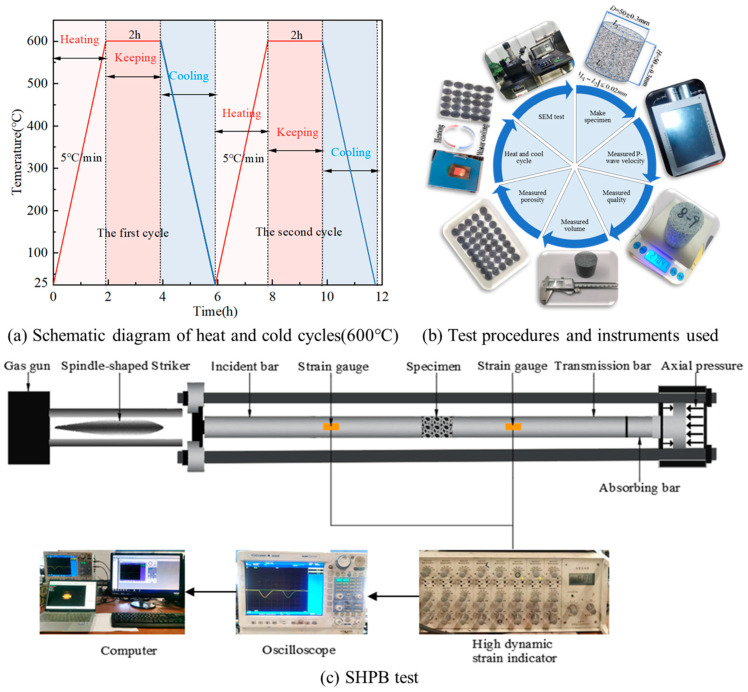
Schematic diagram of the test process and equipment. (**a**) Schematic diagram of heating and cooling cycles (600 °C), (**b**) Test procedures and instruments used, (**c**) SHPB test system.

**Figure 4 materials-16-02300-f004:**
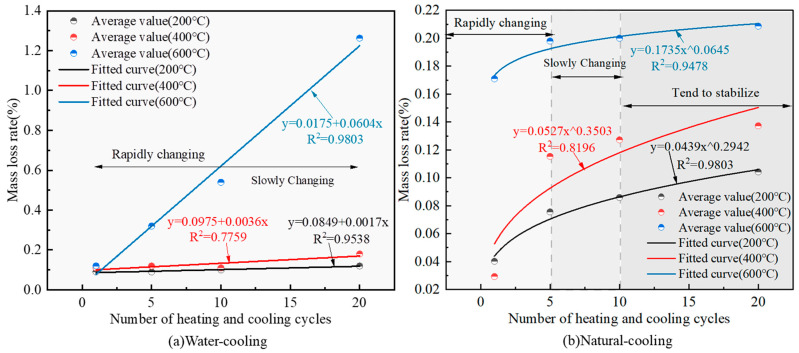
Variation pattern of granite mass change rate under the effect of different heating and cooling cycles.

**Figure 5 materials-16-02300-f005:**
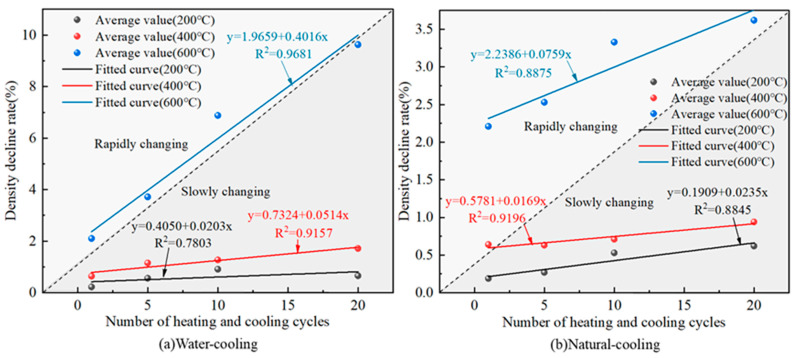
Variation pattern of granite density change rate under different heating and cooling cycles.

**Figure 6 materials-16-02300-f006:**
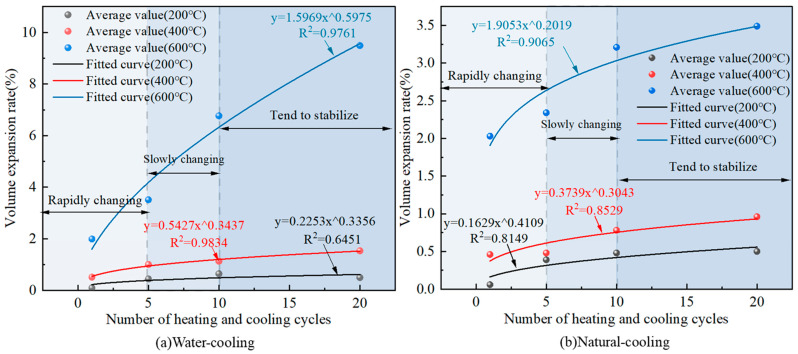
Variation pattern of granite volume change rate under different heating and cooling cycles.

**Figure 7 materials-16-02300-f007:**
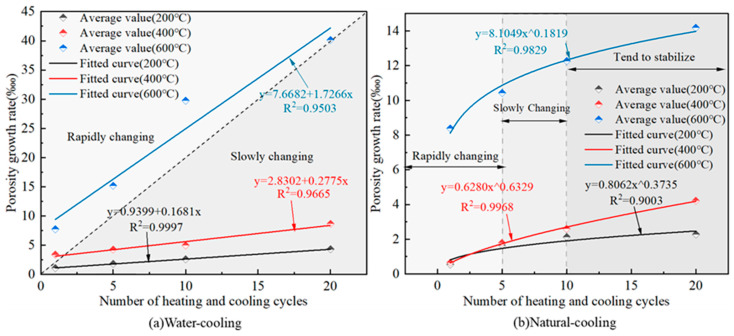
Variation pattern of granite porosity change rate under different heating and cooling cycles.

**Figure 8 materials-16-02300-f008:**
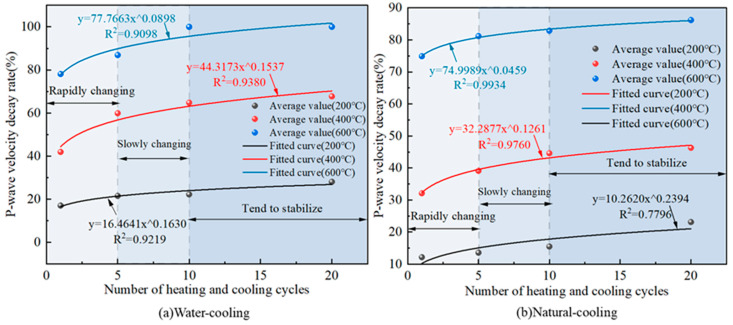
Variation pattern of longitudinal wave velocity change rate of granite under different heating and cooling cycles.

**Figure 9 materials-16-02300-f009:**
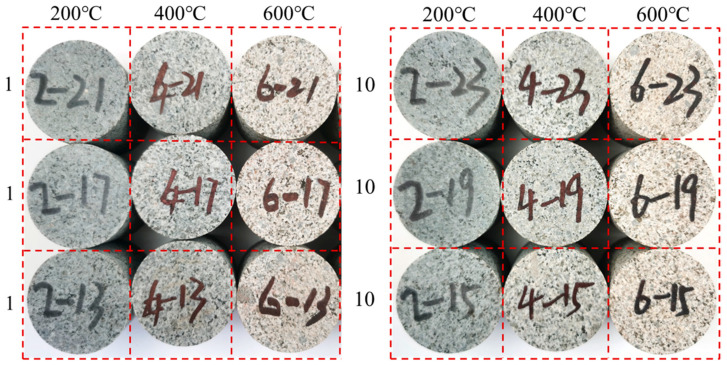
Changes in surface color of granite specimens under the action of different heating and cooling cycles.

**Figure 10 materials-16-02300-f010:**
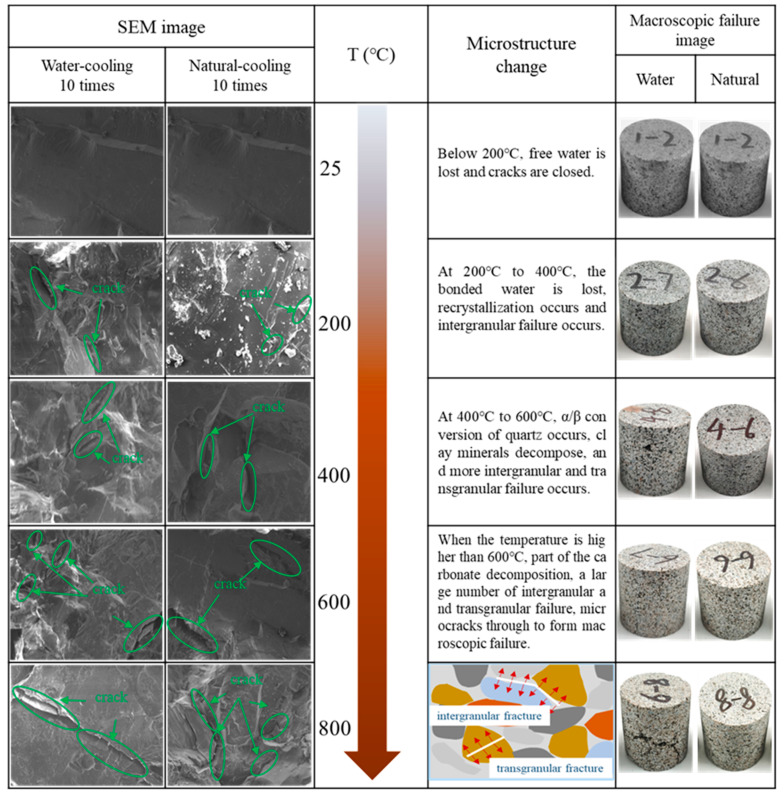
Changes in granite microstructure under different heating and cooling cycles.

**Figure 11 materials-16-02300-f011:**
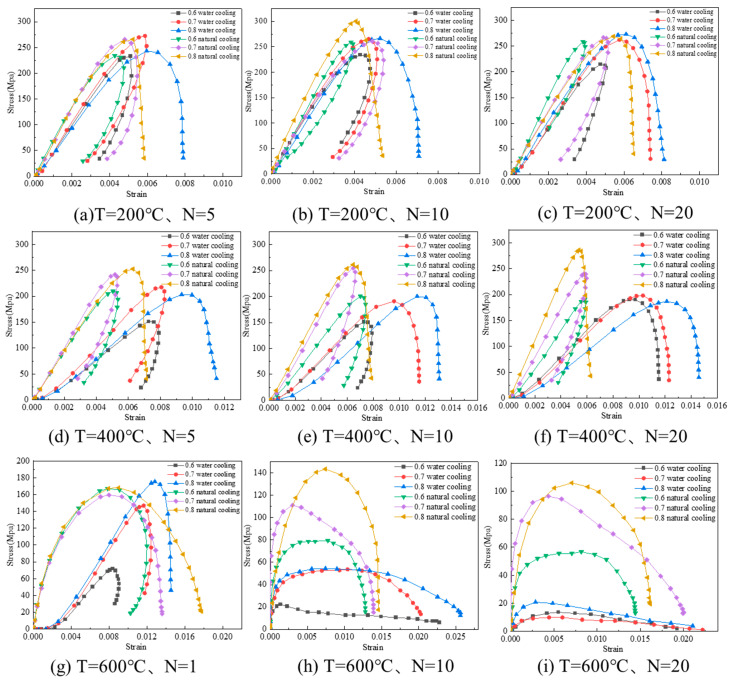
This is a figure. Schemes follow the same formatting.

**Figure 12 materials-16-02300-f012:**
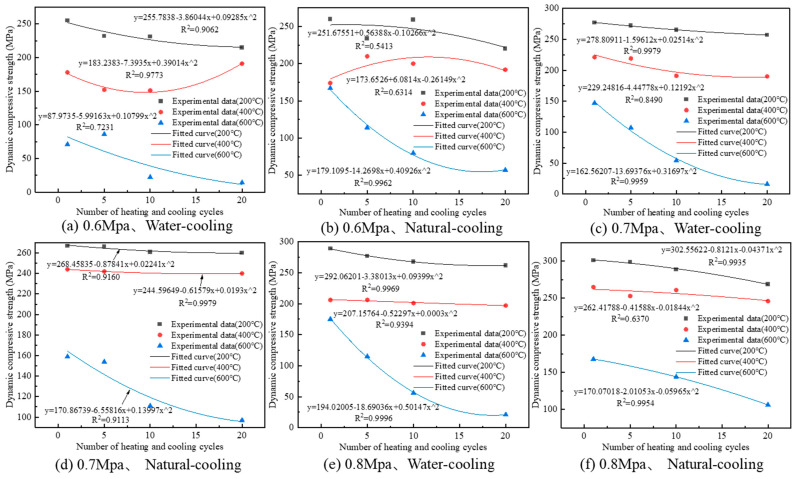
Dynamic compression strength variation pattern of granite specimens after different heating-cooling cycle treatments.

**Figure 13 materials-16-02300-f013:**
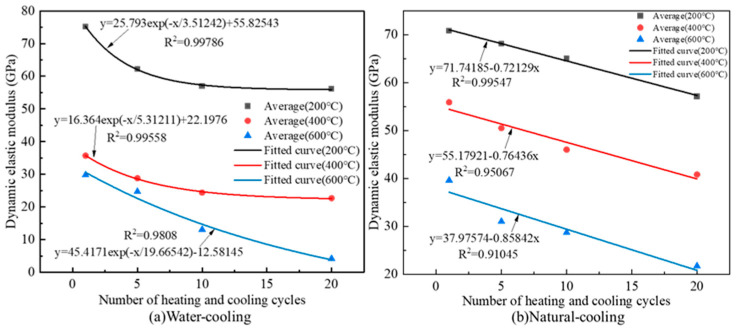
Dynamic elastic modulus changes law of granite specimens after different heating and cooling cycle treatments.

**Figure 14 materials-16-02300-f014:**
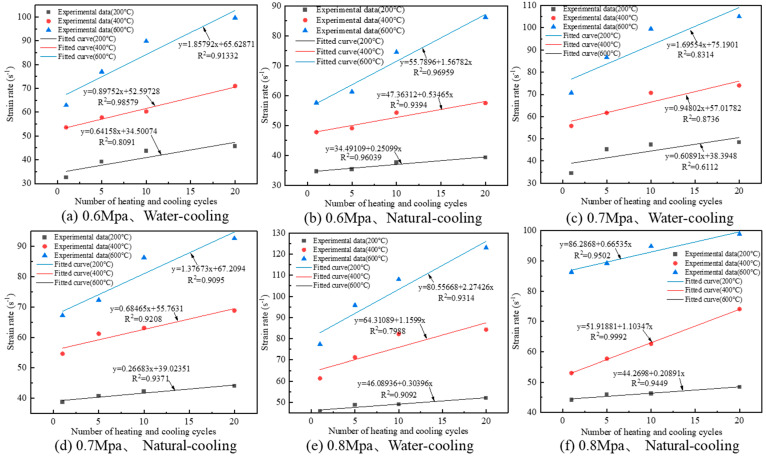
Strain rate variation law of granite specimens after different heating and cooling cycle treatments.

**Figure 15 materials-16-02300-f015:**
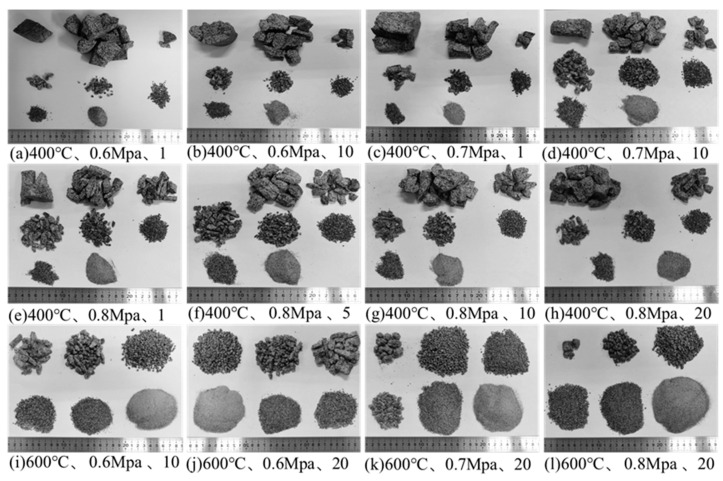
Crushing of granite specimens under different conditions.

**Table 1 materials-16-02300-t001:** Change rate of physical parameters of granite.

T (°C)	Type of Cooling	N	Rate of Change of Physical Parameter				
			$\delta_{m}$/%	$\delta_{v}$/%	$\delta_{\rho }$/%	$\delta_{p}$/%	$\delta_{wv}$/%
200	Water	1	0.09	0.13	0.22	108.9	17.0
		5	0.09	0.45	0.56	180.7	21.5
		10	0.10	0.80	0.91	260.9	22.2
		20	0.12	0.50	0.66	430.3	28.0
	Natural	1	0.04	0.06	0.19	54.2	12.2
		5	0.08	0.39	0.27	164.6	13.6
		10	0.10	0.48	0.53	213.5	15.5
		20	0.10	0.50	0.62	228.7	23.1
400	Water	1	0.11	0.52	0.64	338.5	41.9
		5	0.12	1.01	1.15	427.2	59.9
		10	0.11	1.14	1.27	499.4	64.8
		20	0.18	1.53	1.71	693.7	67.7
	Natural	1	0.03	0.46	0.64	66.5	32.1
		5	0.12	0.48	0.63	179.1	39.1
		10	0.13	0.78	0.71	260.4	44.6
		20	0.14	0.96	0.94	422.5	46.3
600	Water	1	0.12	1.99	2.10	773.9	78.1
		5	0.31	3.51	3.72	1518.4	86.9
		10	0.54	6.77	6.88	2976.4	-
		20	1.01	9.49	9.63	4014.5	-
	Natural	1	0.17	2.03	2.21	838.5	74.9
		5	0.20	2.34	2.53	1042.5	81.2
		10	0.20	3.21	3.33	1226.6	82.9
		20	0.21	3.49	3.62	1419.4	86.2
800	Water	1	0.23	2.51	2.61	1476.2	78.6
	Natural	1	0.21	2.48	2.66	1454.4	75.7
	Furnace	1	0.20	2.02	2.21	1277.7	74.9

**Table 2 materials-16-02300-t002:** Dynamic mechanical parameters of specimens at different conditions.

T (°C)	Type of Cooling	N	Pressure of Impact (Mpa)								
			0.6			0.7			0.8		
			$\sigma_{d}$	$E_{d}$	${\dot{\varepsilon }}_{d}$	$\sigma_{d}$	$E_{d}$	${\dot{\varepsilon }}_{d}$	$\sigma_{d}$	$E_{d}$	${\dot{\varepsilon }}_{d}$
25		1	283	66.5	41.6	288	61.4	42.0	295	64.8	43.8
200	Water	1	255	73.0	32.5	277	77.7	34.5	289	74.9	45.6
		5	232	61.2	39.2	272	60.1	45.2	277	65.4	48.7
		10	231	54.1	43.7	265	58.5	47.4	268	58.3	49.0
		20	215	53.9	45.7	257	54.5	48.4	262	60.0	52.0
	Natural	1	260	71.8	34.7	267	68.3	38.7	301	72.4	44.1
		5	234	65.5	35.4	266	69.0	40.7	299	69.8	45.9
		10	259	63.4	37.6	261	66.0	42.3	289	65.5	46.2
		20	220	54.1	39.3	260	59.2	44.0	269	58.1	48.4
400	Water	1	178	35.5	53.6	221	38.5	55.8	206	33.0	61.3
		5	152	25.9	57.8	219	31.9	61.7	206	28.7	71.2
		10	151	24.3	60.3	191	25.1	70.7	201	23.7	82.2
		20	191	22.1	71.0	190	24.7	74.0	197	21.4	84.3
	Natural	1	174	54.1	47.8	244	58.1	54.6	265	55.4	53.0
		5	210	47.5	49.1	242	54.9	61.2	253	49.2	57.7
		10	200	43.4	54.3	255	47.1	63.1	261	47.4	62.6
		20	192	39.5	57.5	240	42.5	68.8	286	40.5	74.1
600	Water	1	71	32.1	62.9	147	27.8	70.7	175	29.4	77.3
		5	86	22.8	77.0	107	26.6	86.6	115	24.8	95.8
		10	22	14.8	89.8	54	14.0	99.5	56	10.5	108
		20	14	3.4	99.7	16	4.9	105	21	4.2	123
	Natural	1	167	39.1	57.6	159	39.6	67.2	168	40	86.3
		5	114	33.3	61.3	154	28.7	72.3	93	30.9	89.2
		10	80	28.7	74.5	111	28.4	86.3	144	29.1	94.8
		20	57	20.9	86.2	97	21.9	92.6	106	22.2	98.8
800	Water	1	90	20.7	73.3	147	21.5	85.1	140	22.7	96.6
	Natural	1	123	26.8	71.8	154	24.1	81.9	168	29.3	88.4
	Furnace	1	141	21.8	72.5	187	36.1	77.0	188	33.9	86.4

## Data Availability

The data presented in this study are available on request from the corresponding author.

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
