# Peer review of "Study on the Evolution of Physical Parameters and Dynamic Compression Mechanical Properties of Granite after Different Heating and Cooling Cycles"

_materials, 2023, doi:10.3390/ma16062300_

Round 1
Reviewer 1 Report
This manuscript tests the heating and cooling cycles and dynamic impact on the fine-grained granite specimens at 200~600℃ and 1~20 times respectively to analyze the evolution law of basic physical parameters and dynamic compression mechanical properties of fine-grained granite under the effect of heating and cooling cycles.
The materials used in this work were fine-grained granite from Zhangzhou City, Fujian Province. The whole test process is divided into four steps: measurement of physical parameters before treatment, heating and cooling cycle treatment, measurement of physical parameters after treatment, and dynamic compression. Unfortunately, the instruments used in this work were not explained.
The conclusions were supported by the results and discussion. However, it is supposed to be written in paragraphs instead of points.
They used 36 references which are all new and published 5 years back.
This manuscript is well presented, therefore I recommend this manuscript be accepted after minor revision.

Reviewer 2 Report
The review entitled "Study on the evolution of physical parameters and dynamic compression mechanical properties of granite after different heating and cooling cycles” carried out heating and cooling cycles and dynamic impact tests on many fine-grained granite specimens to analyze the evolution law of basic physical parameters and dynamic compression mechanical properties of fine-grained granite.
The manuscript is very well written and provides valuable information and results. However, the following comments should be addressed before any further steps.
Comments:
1- Abstract: The used methodology in this manuscript should be well presented.
2- Line 14: Is the provided range of temperature for the heating and cooling cycles? From Figure 3, the cooling temperature should be 25.
3- Line 16: Why the term "heating and cooling cycles" was repeated two times in this sentence?
4- The format of mentioning or citing references should be corrected.
5- The authors should highlight the novelty of this manuscript in the introduction section. Therefore, the authors should increase their discussion on previous related research and highlight how their study is providing a different approach or adding significantly to what has been done.
6- Lines 106-107: I think the country name should be provided since it is an international journal.
7- The conclusion section should be concise to highlight the main conclusion from this study.
Reviewer 3 Report
The authors have presented an interesting work for building materials engineering, with a careful design of experiments and an extensive experimental campaign. It is a research of interest to professionals in the sector. Only minor changes should be made:
In the abstract it would be useful to include some numerical data and not only descriptive data on what happens after the experiments are carried out.
In the introduction, and throughout the article, reference numbers should be enclosed in square brackets. E.g.; Line 49: Li et al [1],
Line 81: ...deep ground. establised..., check punctuation.
Line 110: reference the standard used.
In the results section, there is a lack of a more in-depth discussion of the results where comparisons with other works are established.
Figure 12 and others are too small to be observed.
The limitations encountered during the course of this study should be included in the conclusions section.
The bibliography is not in format.
